# Dual and Multi-Target Cone-Beam X-ray Luminescence Computed Tomography Based on the DeepCB-XLCT Network

**DOI:** 10.3390/bioengineering11090874

**Published:** 2024-08-28

**Authors:** Tianshuai Liu, Shien Huang, Ruijing Li, Peng Gao, Wangyang Li, Hongbing Lu, Yonghong Song, Junyan Rong

**Affiliations:** 1Biomedical Engineering Department, Fourth Military Medical University, Xi’an 710032, China; liutianshuai91@126.com (T.L.); 18637881307@163.com (S.H.); liruijing1025@163.com (R.L.); xiangzhiwanli@163.com (P.G.); liwy1995@fmmu.edu.cn (W.L.); 2Shaanxi Provincial Key Laboratory of Bioelectromagnetic Detection and Intelligent Perception, Xi’an 710032, China; 3School of Software Engineering, Xi’an Jiaotong University, Xi’an 710049, China

**Keywords:** cone-beam X-ray luminescence computed tomography, deep learning reconstruction, spatial resolution, contrast sensitivity, multi-targets

## Abstract

Background and Objective: Emerging as a hybrid imaging modality, cone-beam X-ray luminescence computed tomography (CB-XLCT) has been developed using X-ray-excitable nanoparticles. In contrast to conventional bio-optical imaging techniques like bioluminescence tomography (BLT) and fluorescence molecular tomography (FMT), CB-XLCT offers the advantage of greater imaging depth while significantly reducing interference from autofluorescence and background fluorescence, owing to its utilization of X-ray-excited nanoparticles. However, due to the intricate excitation process and extensive light scattering within biological tissues, the inverse problem of CB-XLCT is fundamentally ill-conditioned. Methods: An end-to-end three-dimensional deep encoder-decoder network, termed DeepCB-XLCT, is introduced to improve the quality of CB-XLCT reconstructions. This network directly establishes a nonlinear mapping between the distribution of internal X-ray-excitable nanoparticles and the corresponding boundary fluorescent signals. To improve the fidelity of target shape restoration, the structural similarity loss (SSIM) was incorporated into the objective function of the DeepCB-XLCT network. Additionally, a loss term specifically for target regions was introduced to improve the network’s emphasis on the areas of interest. As a result, the inaccuracies in reconstruction caused by the simplified linear model used in conventional methods can be effectively minimized by the proposed DeepCB-XLCT method. Results and Conclusions: Numerical simulations, phantom experiments, and in vivo experiments with two targets were performed, revealing that the DeepCB-XLCT network enhances reconstruction accuracy regarding contrast-to-noise ratio and shape similarity when compared to traditional methods. In addition, the findings from the XLCT tomographic images involving three targets demonstrate its potential for multi-target CB-XLCT imaging.

## 1. Introduction

With the emergence of X-ray-excitable nanoparticles, X-ray luminescence computed tomography (XLCT) has garnered increased attention due to its impressive performance [1,2]. In XLCT, X-ray-excitable nanoparticles serve as imaging probes that emit visible or near-infrared (NIR) light upon X-ray irradiation, detectable by an electron-multiplying charge-coupled device (EMCCD) camera. By addressing an inverse problem with a suitable imaging model for X-ray and light photon transport, it is possible to reconstruct the three-dimensional (3D) distribution of nanoparticles within the scanned object. In contrast to bioluminescence tomography (BLT) and fluorescence molecular tomography (FMT), XLCT achieves greater imaging depth due to the superior penetration and collimation of X-rays. Furthermore, employing X-ray-excited nanoprobes effectively mitigates interference from autofluorescence and background fluorescence, thereby enhancing imaging contrast and resolution [3,4,5,6]. The above advantages have led to the rapid development of XLCT, which has broad application prospects in disease diagnosis, as well as in new drug research and development [7,8].

Since the initial demonstration of XLCT, various imaging geometries have been suggested to enhance spatial and temporal resolution. Cone-beam XLCT (CB-XLCT) accelerates the imaging process considerably compared to narrow-beam and fan-beam systems at the expense of spatial resolution, in which the optical imaging acquisition process can complete within two minutes [9,10]. Due to high scattering of light in biological tissues, the first-order approximation model of the radiative transfer equation (RTE) is currently applied to describe the photon propagation in the forward model of CB-XLCT [11,12,13]. Considering the complicated excitation process of XLCT, the deviation between the approximate linear model of RTE and the true nonlinear photon excitation and propagation could not be avoided. To improve its reconstruction quality, the structured prior knowledge or sparse regularization have been employed to penalize the reconstruction objective function [14,15,16,17], which is mostly solved by statistically iterative algorithms. However, model-based reconstruction may struggle to accurately represent the complex and ill-posed nature of CB-XLCT imaging, constraining its application in in vivo imaging.

In recent years, deep learning techniques have significantly advanced the capabilities of structural and molecular imaging modalities, including low-dose CT [18,19,20,21], bioluminescence tomography (BLT) [22,23], and fluorescence molecular tomography (FMT) [24,25]. By providing a large amount of training data to the neural network, the reconstruction architecture learns how to best utilize the data. Compared to traditional reconstruction methods, deep-learning-based reconstruction can effectively avoid errors caused by inaccurate modeling and ill-posed nature in solving inverse problems, achieving better performance than traditional image reconstruction techniques. However, deep learning technology has yet to be applied to CB-XLCT reconstruction.

In this paper, an end-to-end three-dimensional deep encoder–decoder network (DeepCB-XLCT) is proposed for CB-XLCT reconstruction, in which large datasets are utilized to learn the unknown solution to the inverse problem. The DeepCB-XLCT network is designed to establish a nonlinear mapping from input to output, with the parameters of this mapping continuously studied and adjusted during network training. Based on this method, the inaccuracy arising from constructing the photon propagation model or addressing the ill-posed inverse problem can be effectively mitigated.

The rest of this paper is structured as follows. In Section 2, the conventional forward model and inverse problem of CB-XLCT, the proposed deep neural network for the CB-XLCT imaging model, and the training data and optimization training procedure are described in detail. In Section 3, numerical simulations and phantom experiments are performed for evaluating the performance of the proposed reconstruction approach. The results are presented in Section 4, followed by discussions and conclusions in Section 5.

## 2. Methods

### 2.1. Conventional Forward Model and Inverse Problem of CB-XLCT

For CB-XLCT imaging, nanoparticles within an object emit visible or NIR light upon X-ray irradiation. According to previous studies [26], the quantity of optical photons emitted is directly proportional to both the intensity distribution of the X-rays and the concentration of nanoparticles within the object, expressed as follows:(1)S(r)=ΓX(r)n(r)
where *S*(*r*) represents the emitted light from point *r*, Γ denotes the light yield of the nanoparticles, *n*(*r*) is the concentration of nanoparticles, and *X*(*r*) signifies the intensity of X-rays at position *r*, which can be described using the Lambert–Beer law [14]
(2)$$X\left(r\right)=X\left(r_{0}\right)\exp\{-\int_{r_{0}}^{r}\mu_{t}(\tau )\}$$
where $X(r_{0})$ represent the initial X-ray intensity at positions $r_{0}$, and *μ_t_*(*τ*) shows the X-ray attenuation coefficient, which can be derived from the CT data.

Due to the characteristics of high scattering and low absorbing in the visible and NIR spectral window for biological tissues, the diffusion equation (DE) is used to establish the propagation model of the emitted light [27], which can be expressed as
(3)$$-\nabla \cdot \left[D\left(r\right)\nabla \Phi \left(r\right)\right]+\mu_{a}\left(r\right)\Phi \left(r\right)=S\left(r\right)\left(r\in \Omega \right)$$
where *D*(***r***) is the diffusion coefficient, which can be calculated by $D\left(r\right)=1/[3(\mu_{s}^{\prime }\left(r\right)+\mu_{a}\left(r\right))]$, in which $\mu_{s}^{\prime }\left(r\right)$ is the reduced scattering coefficient, *μ_a_*(***r***) is the absorption coefficient, Φ(***r***) represents the photon fluence, and Ω is the image domain.

To solve the diffusion Equation (3), Robin boundary conditions are typically applied [28,29], and can be expressed as
(4)Φ(r)+2κD(r)[ν∇Φ(r)]=0(r∈∂Ω)
where κ is the boundary mismatch parameter, ν represents the outward unit normal vector on the boundary, and ∂Ω is the boundary of Ω.

Based on the finite element method (FEM), the forward model based on Equations (1)–(4) for the practical application of XLCT can be established as
(5)y=Wx+ς
where **y** denotes the actual fluorescence signals measured on the surface of the imaging object, **W** represents a weight matrix, formulated based on previous studies [2], ς signifies the noise inherent in the system, and **x** denotes the unknown distribution of nanoparticles within the imaging object.

To solve the ill-posed inverse problem of Equation (5), the reconstructing method based on the regularization terms are used in the CB-XLCT reconstruction, which can be expressed as
(6)$$x=\arg\underset{x\geq 0}{\min}{\left\|Wx-y\right\|}_{2}^{2}+\lambda {\left\|x\right\|}_{\beta }$$
where *λ* represents the regularization parameter, and ${\left\|•\right\|}_{\beta }$ (0 < *β* < 2) indicates the norm term. When *β* is equal to 1 and 2, it refers to *L*_1_ regularization and *L*_2_ regularization, respectively.

### 2.2. Deep Neural Network for CB-XLCT

In contrast to traditional methods, CB-XLCT reconstruction utilizing deep neural networks does not focus on explicitly solving the forward and inverse problems. Rather, it creates an end-to-end deep neural network mapping model that directly reconstructs the distribution of fluorescence sources. The end-to-end three-dimensional deep encoder–decoder network (DeepCB-XLCT) is composed of a 3D-Encoder network and a 3D-Decoder network, featuring multiple layers of spatial convolution and deconvolution operations. The input to the network comprises 2D surface measurement images from 24 angles of CB-XLCT, while the output is the reconstructed 3D distribution of CB-XLCT nanoparticles. The details of the network structure proposed in this work are shown in Figure 1. The fundamental architecture of the 3D Encoder–Decoder network is divided into two components: the 3D-Encoder network, which captures the distribution of optical features and other codes, and the 3D-Decoder network, which learns the conditional distribution of **y** in order to generate a specific **x**. The 3D-Encoder network is composed of five convolutional blocks. Each block includes a 3D convolutional layer, followed by batch normalization and LeakyReLU activation function, as well as a max pooling layer to progressively reduce spatial dimensions. The 3D-Decoder network uses transposed convolutional layers to progressively upscale the feature maps while reducing the number of channels, consisting of five layers. Each transposed convolutional layer is typically followed by batch normalization and a ReLU activation function, enhancing the network’s non-linear expressive capacity and stability. There are two fully connected layers between the Encoder network and Decoder network, which transform the encoded features to another feature space and then back again, preparing for feature fusion in the decoder. Moreover, the dual-sampling module [24] is integrated at various stages of the 3D-Decoder network, which makes the DeepCB-XLCT more focused on effective features when processing volumetric data, enhancing feature representation and overall network performance. 

### 2.3. Generation and Preparation of Training Data 

In order to obtain a large amount of training data, the forward model of CB-XLCT is solved using computer simulation programs, in which the intensity distribution of the X-rays (Equation (2)) and the number of optical photons emitted (Equation (1)) were solved through analytical calculation, and the propagation model of the emitted light (Equation (3)) was solved based on the finite element method (FEM) [5,14]. To ensure consistency between the simulated data and the actual experiments, the finite element meshes were created based on the phantom or mouse model utilized in the real experiment.

In the phantom experiments, the training data were generated by randomly introducing two or three cylindrical fluorescent targets of varying sizes and locations. These targets, with diameters of either 3 or 4 mm and a fixed height of 4 mm, were distributed randomly throughout the phantoms. The horizontal and vertical coordinates of these fluorescent target centers were randomly generated within the range of −10 to 10 mm, while the edge-to-edge distances (EEDs) ranged from 0.3 to 2.5 mm. For the mouse experiments, training data were created by adding two cylindrical fluorescent targets of different sizes and positions at random. Similarly, cylindrical fluorescent targets with diameters of 3 or 4 mm and a fixed height of 4 mm were randomly positioned inside the mouse model, with their center coordinates generated between −8 and 8 mm to conform to the specifications of the mouse model. The X-ray source and EMCCD configuration of the data were adjusted to replicate the conditions of real experiments. Finally, in the phantom experiments, a total of 8000 simulation samples were generated, with 6000 samples allocated for training and 2000 samples reserved for validation to optimize the model. Similarly, the mouse experiments involved generating 4000 simulation samples, with 3000 samples used for training and 1000 samples set aside for validation. 

Furthermore, all projection images were resized to 128 × 128 pixels prior to being input into the network. The network’s output is a 3D image, with each x-y slice measuring 128 × 128 pixels and a reconstruction resolution of 1 mm along the z-axis. To summarize, the input shape is 128 × 128 × 24 (as illustrated in Figure 1) for the projection images, while the output shape is 128 × 128 × 7 for the reconstructed tomographic images.

### 2.4. Optimization Training Procedure of DeepCB-XLCT

The DeepCB-XLCT networks were implemented using PyTorch 1.12 and Python 3.9 on an Ubuntu 18.04.3 system. All training and testing procedures were conducted using two NVIDIA-A100 with 80 GB memory for each. The objective function of the DeepCB-XLCT network consists of two parts: the mean square errors (MSE) and structural similarity loss (SSIM) between the output of DeepCB-XLCT and true results. In addition, due to the smaller proportion of the target region compared to the entire reconstruction area as shown in Figure 2, the loss of the target regions (ROI) is also considered. 

The Structural Similarity Index (*SSIM*) [30] is a perceptually motivated metric that evaluates the similarity between two images by comparing corresponding pixels and their surrounding neighborhoods. Specifically, the SSIM index employs a sliding Gaussian window to assess the similarity of each local patch within the images. The SSIM of the same pixels in the output of DeepCB-XLCT and true results is defined as
(7)$$SSIM(x_{r_{i}},x_{t_{i}})=\frac{\left(2\mu_{x_{r_{i}}}\mu_{x_{t_{i}}}+C_{1})(2\sigma_{x_{r_{i}}x_{t_{i}}}+C_{2}\right)}{\left(\mu_{x_{r_{i}}}^{2}+\mu_{x_{t_{i}}}^{2}+C_{1})(\sigma_{x_{r_{i}}}^{2}+\sigma_{x_{t_{i}}}^{2}+C_{2}\right)}$$
where $\mu_{x_{r_{i}}}$ and $\mu_{x_{t_{i}}}$ are the means of pixel *i* in the output of DeepCB-XLCT and true results, respectively, $\sigma_{x_{r_{i}}}$ and $\sigma_{x_{t_{i}}}$ are the variances of of pixel *i*. *C*_1_ and *C*_2_ are constants used to stabilize the divisions. *x_r_* is the output of DeepCB-XLCT, and *x_t_* is the true results.

To compute the *SSIM* index between the output of DeepCB-XLCT and the true results, the mean is taken over the *SSIM* indices of all local patches.
(8)$$MSSIM(x_{r},x_{t})=\frac{1}{M}\sum_{i=1}^{M}SSIM(x_{r_{i}},x_{t_{i}})$$
where *M* is the number of local patches in the output of DeepCB-XLCT and true results.

Then, the *SSIM* loss between the output of DeepCB-XLCT and true results can be defined as
(9)$${ℒ}^{SSIM}(x_{r},x_{t})=1-MSSIM(x_{r},x_{t})$$

In the end, the objective function of the DeepCB-XLCT network can be defined as
(10)$${ℒ}_{DeepCB-XLCT}={ℒ}^{MSE}(x_{r},x_{t})+{ℒ}^{MSE}(x_{r}^{\prime },x_{t}^{\prime })+2\cdot [{ℒ}^{SSIM}(x_{r},x_{t})+{ℒ}^{SSIM}(x_{r}^{\prime },x_{t}^{\prime })]$$
where $x_{r}^{\prime }$ and $x_{t}^{\prime }$ are the target regions in the output of DeepCB-XLCT and true results, respectively. The *MSE* loss between the output of DeepCB-XLCT and the true results can be calculated as
(11)$${ℒ}^{MSE}(x_{r},x_{t})=\frac{1}{N}\sum_{i=1}^{N}\left(x_{r_{i}}-x_{t_{i}}\right)$$
where *N* is the number of training samples.

In this study, the network utilized the Adam algorithm as its optimizer. The batch size was 64, which denoted the number of samples input to the model at once during the training process. The epochs ware set to 200 based on the tradeoff between enough training and avoiding overfitting. The initial learning rate is 2 × 10^−5^, and the learning rate decays based on the validation loss value, which the learning rate decreases by 2 times when the validation loss remains unchanged for 5 cycles. Under these parameter conditions, it takes 3 h to train the DeepCB-XLCT model.

## 3. Experimental Design

Numerical simulations and phantom experiments were conducted to assess the performance of the proposed DeepCB-XLCT network in CB-XLCT reconstruction using the custom-developed CB-XLCT system. Additionally, for comparison, four traditional methods, namely, adaptive FISTA (ADFISTA, *L*_1_ norm), MAP (Gaussian model) [14,31,32], T-FISTA (*L*_1_ norm) [33], and ADMLEM (Poisson model) [2], were implemented to solve Equation (6). In this study, the gradient variation step of the ADFISTA method was set as 0.01, and the iteration numbers were determined by an adaptive iteration termination condition. The MAP method was established based on Gaussian Markov random field, in which the hyperparameters were alternately estimated in each iteration and the objective function was minimized based on a voxel-wise iterative coordinate descent (ICD) algorithm. The regularization parameters and iterative numbers of the T-FISTA method were empirically set as 0.1 and 300, respectively. The ADMLEM method was established based on the Poisson distributed projections, in which the iterative number was set as 800.

### 3.1. Numerical Simulations Setup

Numerical simulations were first conducted using a cylinder phantom to evaluate the performance of the proposed DeepCB-XLCT network. To simulate the environment of biological tissues, the cylindrical phantom, measuring 3.0 cm in diameter and 2.3 cm in height, was filled with a mixture of water and intralipid, as shown in Figure 3. The absorption coefficient and reduced scattering coefficient were specified as 0.02 and 10 cm^−1^, respectively. Additionally, two small tubes (4 mm in diameter and 4 mm in height) containing Y_2_O_3_:Eu^3+^ (50 mg/mL) were positioned within the cylinder phantom as the luminescent targets. To assess the resolution capabilities of the proposed DeepCB-XLCT network, the edge-to-edge distances (EEDs) between the two targets were set as 2.0, 1.5, and 1.0 mm.

In the numerical simulations, 24 projections were obtained every 15° during a 360° scan, with the voltage and current of the cone beam X-ray source set at 40 kV and 1 mA, respectively. The exposure time of the EMCCD camera was fixed at 0.5 s. Following acquisition of optical luminescence data from various angles, white Gaussian noise with a zero-mean and a signal-to-noise ratio (SNR) of 30 dB was added to all projections to simulate noisy measurements. Moreover, in order to further validate the performance of the proposed DeepCB-XLCT method under varying noise levels, the projections were also added with 25 dB, 20 dB, and 15 dB white Gaussian noise. 

### 3.2. Phantom Experiments Setup

To further assess the performance of the proposed DeepCB-XLCT network using actual luminescence measurements, phantom experiments were conducted with a custom-developed CB-XLCT system, which mainly included a rotation stage, a micro-focus X-ray source (Oxford Instrument, Oxford, UK), an electron-multiplying charge-coupled device (EMCCD) camera (iXon DU-897, Andor, Oxford, UK) for capturing luminescence signals, and a flat-panel X-ray detector (2923, Dexela, London, UK) for capturing X-ray signals, as depicted in Figure 4. The X-ray source operated at a maximum voltage of 80 kV with a maximum power of 80 W.

Figure 5 illustrates the setup of the physical phantom utilized in the imaging experiments. A glass cylinder measuring 4.0 cm in diameter and 4.0 cm in height, filled with a mixture of 1% intralipid and water, was secured on the rotation stage. The absorption coefficient and reduced scattering coefficient of the solution were 0.02 cm^−1^ and 10 cm^−1^, respectively. Additionally, two small glass tubes (3 mm in diameter) containing Y_2_O_3_:Eu^3+^ (50 mg/mL) were symmetrically positioned within the cylinder as the luminescent targets, with edge-to-edge distances between the two tubes of 2.3 mm, 1.7 mm, and 1.0 mm.

In the phantom experiments, the X-ray source was configured with a tube voltage of 40 kV and a tube current of 1 mA. Optical images were captured at every 15° as the phantom rotated from 0° to 360° using the EMCCD camera. The exposure time, EM gain, and binning of the EMCCD were configured to 0.5 s, 260, and 1 × 1, respectively. For CT imaging, 360 projections were acquired with an angular increment of 1° spanning from 0° to 360° by the X-ray detector, with each projection lasting 150 ms. Subsequently, the Feldkamp–Davis–Kress (FDK) algorithm [34,35] was employed to reconstruct the conventional CT image of the tubes for validation.

### 3.3. Quantitative Evaluation

The CB-XLCT images reconstructed using ADFISTA, MAP, T-FISTA, ADMLEM, and the proposed DeepCB-XLCT network were evaluated and compared based on quantitative metrics, namely, the Dice Similarity Coefficient (*DICE*) and Contrast-to-Noise Ratio (*CNR*) [12].

The *DICE* coefficient is employed to assess the similarity between the true and reconstructed targets, and it can be calculated using the following formula:(12)$$DICE=\frac{2\left|ROI_{r}\cap ROI_{t}\right|}{\left|ROI_{r}\right|+\left|ROI_{t}\right|}$$
where *ROI_t_* and *ROI_r_* represent the regions corresponding to the true and reconstructed targets, respectively, and |·| indicates the number of voxels within a given region.

The *CNR* is utilized for the quantitative assessment of contrast and noise in the reconstructed images, as outlined in Equation (13).
(13)$$CNR=\frac{\left|\mu_{ROI}-\mu_{BCK}\right|}{{\left(w_{ROI}\sigma_{ROI}^{2}+w_{BCK}\sigma_{BCK}^{2}\right)}^{1/2}}$$
where *ROI* and *BCK* represent the target and background regions, respectively, and $\mu_{ROI}$ and $\sigma_{ROI}^{2}$ are the mean intensity values and the variances within the *ROI*
$\mu_{BCK}$ and $\sigma_{BCK}^{2}$ represent the mean intensity values and variances of the *BCK*, and $w_{ROI}$ and $w_{BCK}$ serve as weighting factors based on the relative volumes of the *ROI* and *BCK*, respectively, which satisfy $w_{ROI}$ + $w_{BCK}$ = 1.

## 4. Results

### 4.1. Numerical Simulations

#### 4.1.1. Resolution Experiment

Firstly, CB-XLCT reconstruction results were obtained for targets placed at varying edge-to-edge distances (EEDs) using ADFISTA, MAP, T-FISTA, ADMLEM, and the proposed DeepCB-XLCT network, as depicted in Figure 6. Normalization of all reconstruction results was performed based on their respective maximum intensity values. It can be observed that in the cases of the ADFISTA, MAP, and T-FISTA algorithms, distinguishing between the two targets proved challenging. With the ADMLEM method, while the distribution of the two targets could be differentiated, it exhibited relatively large localization errors. In contrast, the reconstruction using the proposed DeepCB-XLCT network demonstrated improved shape similarity to the ground truth and resulted in less noise compared to the other four algorithms.

Table 1 provides a comprehensive overview of the quantitative assessment of reconstructions employing various methods. Notably, the results obtained using the proposed DeepCB-XLCT network demonstrated the highest reconstruction quality in terms of *CNR* and shape similarity among all five methods, thereby reinforcing the observations depicted in Figure 6. Figure 7 plots the profiles along the green dotted line shown in Figure 6. The results again reveal that the values of the reconstructed images by the proposed DeepCB-XLCT network are closer to the ground truth images.

#### 4.1.2. Robustness Experiment in Different Noise Levels and the Ablation Experiment

To further assess the performance of the proposed method across varying noise levels, XLCT tomographic images from two-target simulations with SNR values of 25 dB, 20 dB, and 15 dB were reconstructed, as illustrated in Figure 8. The quantitative evaluation of these reconstructions is summarized in Table 2. It is evident that the reconstruction results obtained using the proposed DeepCB-XLCT network maintained an acceptable quality concerning *CNR* and shape similarity, even when the SNR decreased to 15 dB. This finding further substantiates the enhancements provided by the proposed algorithm.

To investigate the impact of skip connection module and ROILoss on the reconstruction results, the ablation experiments were conducted based on the targets positioned at varying distances, as shown in Figure 9. Table 3 provides a summary of the quantitative evaluation of the reconstructions in ablation experiment. It can be seen without skip connection module and ROILoss, the quality of reconstruction results significantly decreased. Compared with skip connection module, ROILoss had a greater impact on the reconstruction results.

#### 4.1.3. Multi-Target Experiment

Subsequently, XLCT tomographic images of three targets were reconstructed using various algorithms to assess the effectiveness of the proposed method in the context of multi-target scenarios, as illustrated in Figure 10. Notably, none of the algorithms, except for the proposed DeepCB-XLCT network, could effectively distinguish between the targets, which further confirms the enhancements brought by the proposed algorithm for XLCT tomographic images involving multiple targets. 

### 4.2. Phantom Experiments

To further assess the efficacy of the proposed DeepCB-XLCT network, XLCT tomographic images of targets with varying EEDs were reconstructed from phantom experiments using different algorithms, as depicted in Figure 11. It is evident that, for ADFISTA and MAP algorithms shown in the first and second columns of Figure 11, distinguishing the distribution of the two targets proves to be challenging. Comparatively, while the T-FISTA and ADMLEM algorithms were able to effectively distinguish between the distributions of two targets at EEDs of 2.3 mm and 1.7 mm, they struggled to do so when the EED was 1.0 mm. Notably, the proposed DeepCB-XLCT network successfully resolved both targets as expected, as illustrated in the fifth column of Figure 11.

To further assess the proposed algorithm, Table 4 provides a quantitative evaluation of the reconstructions obtained through various methods in the phantom experiments. The results demonstrate that, in comparison to conventional reconstruction methods, the proposed DeepCB-XLCT network consistently outperforms in terms of shape recovery and image contrast in phantom experiments, which further corroborates the advancements made by the proposed DeepCB-XLCT network.

### 4.3. In Vivo Experiments and Results

To substantiate the efficacy of the proposed method in vivo, CB-XLCT experiments were performed on a female BALB/c nude mouse using our custom-developed CB-XLCT system. All procedures adhered strictly to the guidelines established by the Animal Ethics Review Committee of the Fourth Military Medical University. Two small glass tubes, measuring 3 mm in diameter and 4 mm in height, filled with Y_2_O_3_:Eu^3+^ at a concentration of 50 mg/mL, were implanted into the mouse’s abdominal cavity to function as fluorescent targets, as illustrated in Figure 12a,b. During the imaging experiments, the mouse was secured on the rotation stage, and 24 projections were acquired at 15° intervals from 0° to 360° by the EMCCD camera, while the X-ray source operated at 80 kV and 1 mA, respectively. The exposure time and EM gain of the EMCCD camera were set as 1 s and 260, respectively. Then, the CT projections were also collected to reconstruct the localization and structure of the tubes for verification, as shown in Figure 12b.

The XLCT tomographic images were reconstructed using ADFISTA, MAP, T-FISTA, ADMLEM, and the proposed DeepCB-XLCT method, as illustrated in Figure 12c–l. All fluorescence images were normalized according to the maximum reconstruction results and presented using a uniform color scale, as depicted in Figure 12c–g. Figure 12h–l presents the fusion outcomes of the XLCT and XCT reconstructed images. In comparison to the ADFISTA, MAP, T-FISTA, and ADMLEM method, the proposed DeepCB-XLCT method exhibits enhanced reconstruction performance in terms of image contrast and shape recovery.

Table 5 provides a quantitative evaluation summary of the reconstructions obtained using various methods. The proposed DeepCB-XLCT achieved the highest values for *CNR* and *DICE*, indicating its capability to produce high-quality reconstruction results in mouse experiments. Table 6 summarizes the reconstruction time of ADFISTA, MAP, T-FISTA, ADMLEM, and the proposed DeepCB-XLCT method in the mice experiments. Compared with the conventional iterative methods, the reconstruction time of the proposed DeepCB-XLCT method is significantly reduced due to the absence of complex iterative calculations.

## 5. Discussion and Conclusions

CB-XLCT, as a novel hybrid imaging modality, offers significant advantages primarily due to its use of X-rays, which can enhance excitation depth, reduce tissue autofluorescence, and achieve a dual-mode imaging capability that combines X-ray computed tomography (CT) with optical molecular tomographic imaging. The performance of CB-XLCT reconstruction has a substantial effect on imaging outcomes. Nevertheless, the deviation between the complex imaging process and the approximate photon transmission model leads to the high ill-posed nature of the inverse problem, which greatly limits the improvement of imaging quality.

Conventionally, researchers have developed various reconstruction methods to constrain the image and improve the reconstruction quality, including ADFISTA, Bayesian theory, T-FISTA, and ADMLEM algorithms to mitigate the ill-posed nature of the inverse problem. In contrast to these conventional methods, this study introduces a novel DeepCB-XLCT approach, in which the CB-XLCT reconstruction process is completed by establishing an end-to-end nonlinear mapping model that relates the internal CB-XLCT nanoparticles to the surface measurement signals. Based on the DeepCB-XLCT method, it can greatly eliminate the modeling error associated with the forward problem and effectively minimize artifacts generated by iterative calculations, thereby greatly addressing the challenges posed by the ill-posed inverse problem. Moreover, the structural similarity loss (SSIM) was added to the objective function of the DeepCB-XLCT network to better restore the target shape, and the loss of the target regions was also added to make the DeepCB-XLCT network more focused on the target, which were validated based on the ablation experiment. Finally, it is worth mentioning that a certain level of noise in the data will not compromise the robustness of the DeepCB-XLCT method. In summary, the proposed DeepCB-XLCT method can achieve improved reconstruction results and robust reconstruction performance.

Numerical simulations, phantom experiments, and in vivo experiments results validate the superiority of the proposed method compared to traditional approaches such as the ADFISTA, MAP, T-FISTA, and ADMLEM methods. When the EED is 1.0 mm, two targets can be distinctly resolved (Figure 6 and Figure 11, Table 1 and Table 4), highlighting its effectiveness in enhancing spatial resolution. The reconstruction results in different noise levels (Figure 8, Table 2) further confirmed the robustness of the proposed DeepCB-XLCT method. The results of the ablation experiment (Figure 9, Table 3) indicate that the ROILoss has a greater influence on the reconstruction outcomes. Moreover, the results from the XLCT tomographic images with three targets (Figure 10) reveal the method’s capability for multi-target CB-XLCT imaging. To further assess the effectiveness of the proposed method, in vivo experiments were performed, and the results (Figure 12, Table 5) provide additional confirmation of its superiority over the traditional ADFISTA, MAP, T-FISTA, and ADMLEM methods, which would significantly enhance the preclinical application of CB-XLCT in small animals.

However, there are still some limitations. Firstly, the performance of the proposed DeepCB-XLCT method has been validated through phantom experiments with varying EEDs. To fully demonstrate the method’s superior capabilities, it is better to verify its performance through in vivo experiments involving different targets in the future. Secondly, as the number of reconstruction targets increase, the complexity of network design and training may also increase. Moreover, while a well-trained model can reconstruct the distribution of XLCT nanoparticles in a very short time, the training process for the network can be time-consuming, taking several hours. In the future, we will concentrate on addressing these challenges and advancing our research in several areas, including the development of improved network structures, conducting evaluations through in vivo experiments, and testing the robustness of the proposed method using various phantom and in vivo studies.

In summary, we have proposed a DeepCB-XLCT method to enhance the quality of CB-XLCT reconstruction. It directly constructs a nonlinear mapping relationship between the distribution of X-ray excitable nanoparticles and the boundary fluorescent signal distribution, which could effectively reduce the reconstruction inaccuracies associated with simplified linear models and iterative calculations. The results from numerical simulations, phantom experiments, and in vivo experiments have demonstrated that the proposed DeepCB-XLCT method can improve spatial resolution and reconstruction accuracy compared to conventional iterative methods, which can advance the preclinical application of CB-XLCT in small animals.

## Figures and Tables

**Figure 1 bioengineering-11-00874-f001:**
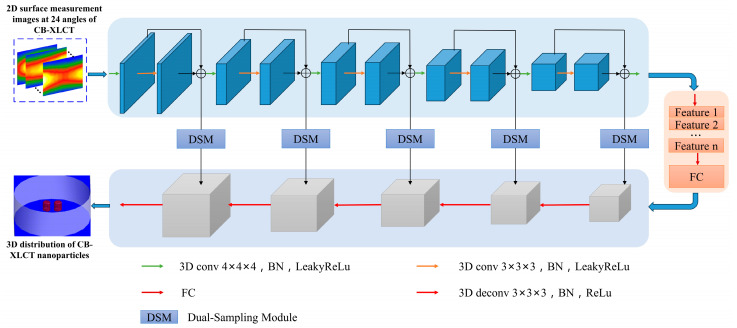
Schematic illustration of the DeepCB-XLCT network architecture: the 3D deep encoder–decoder (3D-En–Decoder) network has a 3D-Encoder network and a 3D-Decoder network. The 3D encoder consists of several convolution layers, followed by batch norm, ReLU activation function, and pooling. The 3D-Decoder has several upsampling layers followed by convolution, batch norm, and ReLU activation function. There are two full connection layers between the 3D-Encoder and the 3D-Decoder.

**Figure 2 bioengineering-11-00874-f002:**
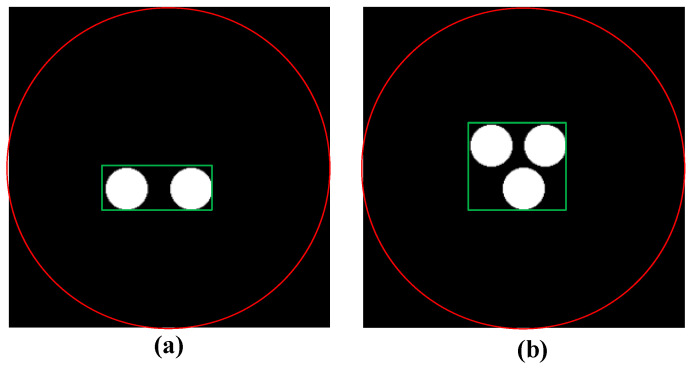
The cylinder phantom used in the training data simulation with two targets and three targets (**a**) two targets, (**b**) three targets, the region within the red circle is the entire reconstruction region, the region within the green rectangular box is the target region.

**Figure 3 bioengineering-11-00874-f003:**
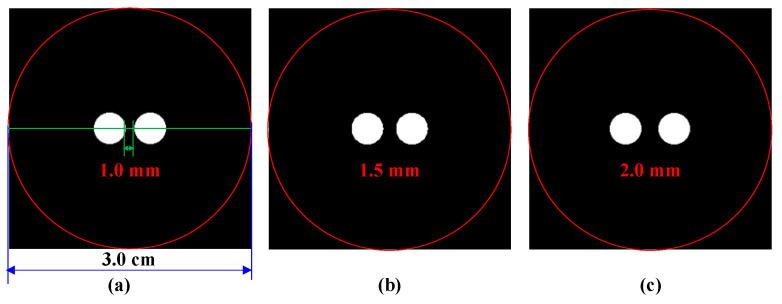
The cylinder phantom used in simulation studies. Edge-to-edge distance between the two targets: (**a**) 1 mm, (**b**) 1.5 mm, (**c**) 2 mm, the red circle is the boundary of the phantom, the green line is the centerline of the phantom.

**Figure 4 bioengineering-11-00874-f004:**
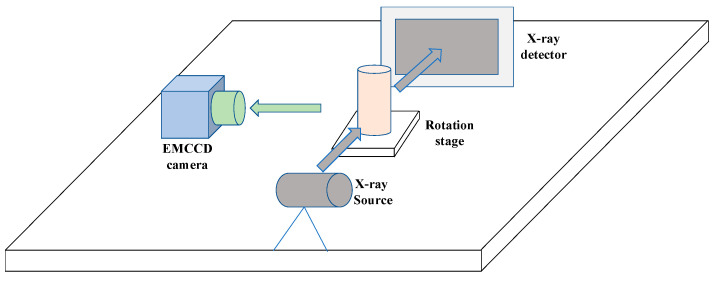
The schematic diagram of the CB-XLCT system.

**Figure 5 bioengineering-11-00874-f005:**
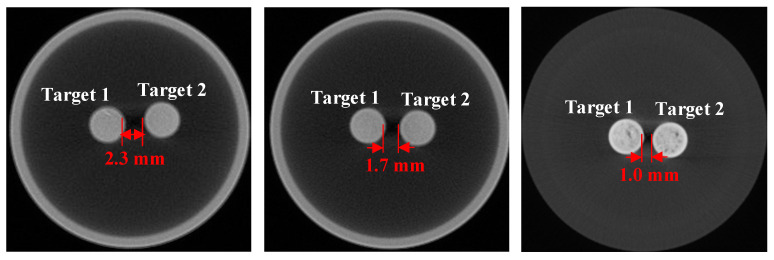
Configuration for the two-target phantom experiment. The concentrations of the two targets were both 50 mg/mL, and the edge-to-edge distances of the two targets were 2.3 mm, 1.7 mm, and 1.0 mm.

**Figure 6 bioengineering-11-00874-f006:**
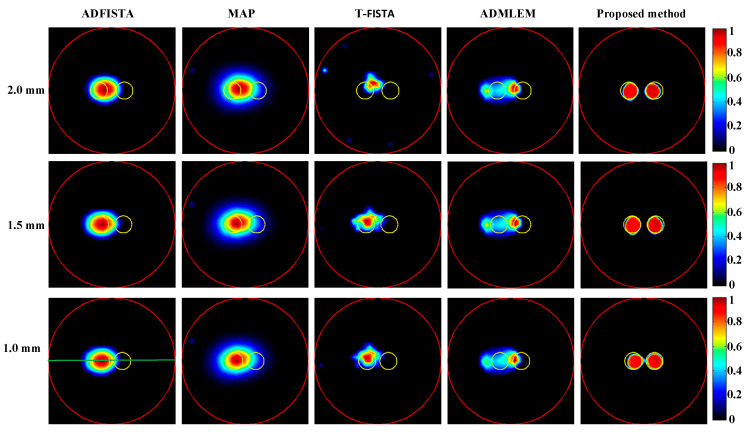
Tomographic images of the targets positioned at varying distances were reconstructed. The first through fifth columns present the results obtained using ADFISTA, MAP, T-FISTA, ADMLEM, and the proposed DeepCB-XLCT network, respectively. The first to third rows correspond to reconstructions with edge-to-edge distances of 2.0 mm, 1.5 mm, and 1.0 mm, respectively. The red circle is the boundary of the phantom, the green line is the centerline of the reconstruction image and the yellow circle represents the true target region.

**Figure 7 bioengineering-11-00874-f007:**
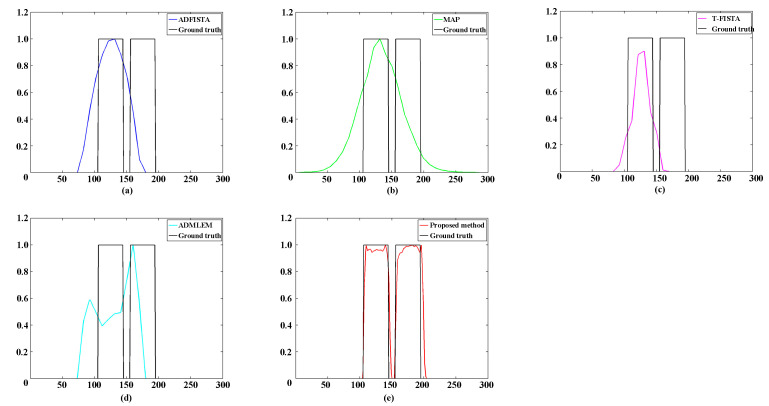
Profiles along the green dotted line in Figure 5 with the edge-to-edge distances of 1.0 mm. (**a**–**e**) The profiles of reconstruction results achieved with ADFISTA, MAP, T-FISTA, ADMLEM, and the proposed DeepCB-XLCT network, respectively.

**Figure 8 bioengineering-11-00874-f008:**
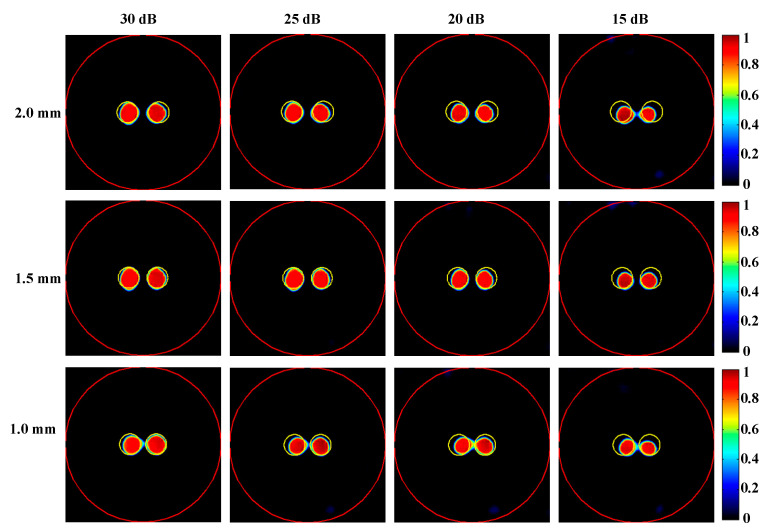
Tomographic images of the targets positioned at different distances with different noise levels were reconstructed based on the proposed DeepCB-XLCT network. The red circle is the boundary of the phantom and the yellow circle represents the true target region. The first to fourth columns are the reconstruction results with the SNRs of 30 dB, 25 dB, 20 dB, and 15 dB, respectively. The first to third row are the results reconstructed with the edge-to-edge distances of 2.0 mm, 1.5 mm, and 1.0 mm, respectively.

**Figure 9 bioengineering-11-00874-f009:**
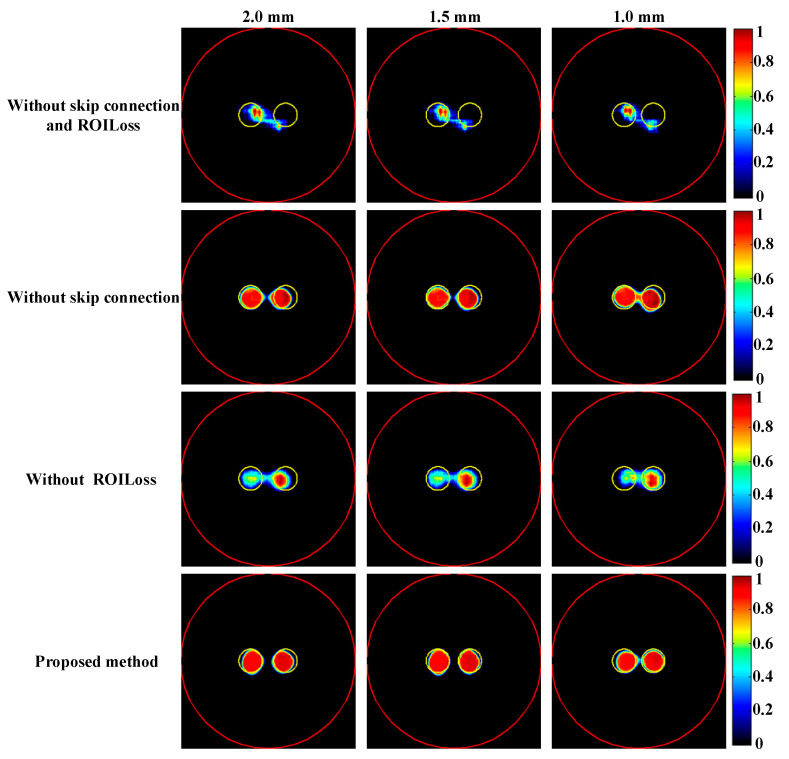
Tomographic images of the targets positioned at different distances were reconstructed in the ablation experiment. The red circle is the boundary of the phantom and the yellow circle represents the true target region. The first to fourth rows are the results obtained based on the proposed DeepCB-XLCT network without skip connection and ROILoss, the proposed DeepCB-XLCT network without skip connection, the proposed DeepCB-XLCT network without ROILoss, and the proposed DeepCB-XLCT network, respectively. The first to third columns are the results reconstructed with the edge-to-edge distances of 2.0 mm, 1.5 mm, and 1.0 mm, respectively.

**Figure 10 bioengineering-11-00874-f010:**
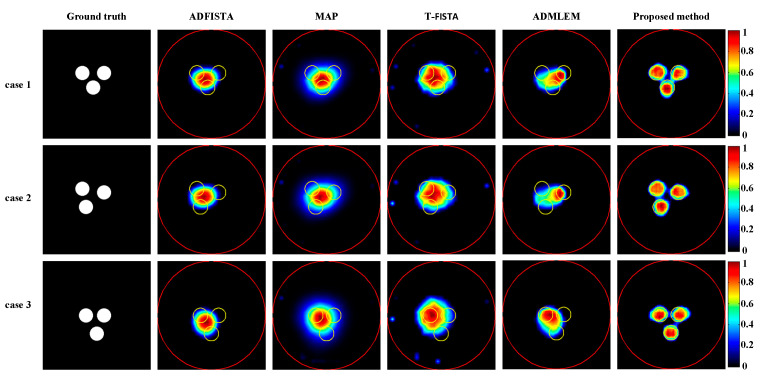
Tomographic images of three targets were reconstructed using various methods. The red circle is the boundary of the phantom and the yellow circle represents the true target region. The first column depicts the true locations of the targets. The second to sixth columns present results obtained using ADFISTA, MAP, T-FISTA, ADMLEM, and the proposed DeepCB-XLCT network, respectively. The first to third rows show the reconstructions for the three targets positioned at different locations.

**Figure 11 bioengineering-11-00874-f011:**
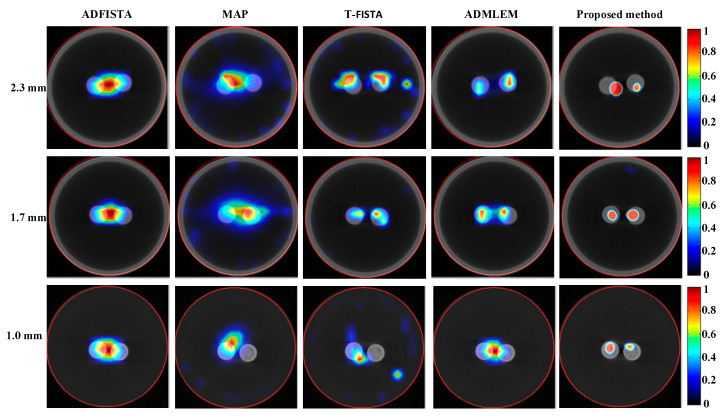
Tomographic images of the targets positioned at varying distances were reconstructed based on different algorithms for phantom experiments. The first to third rows display the fused XLCT/CT tomographic images for the EEDs of 2.3, 1.7, and 1.0 mm, respectively. The reconstructions obtained from ADFISTA, MAP, T-FISTA, ADMLEM, and the proposed DeepCB-XLCT network are presented in the first through fifth columns, respectively.

**Figure 12 bioengineering-11-00874-f012:**
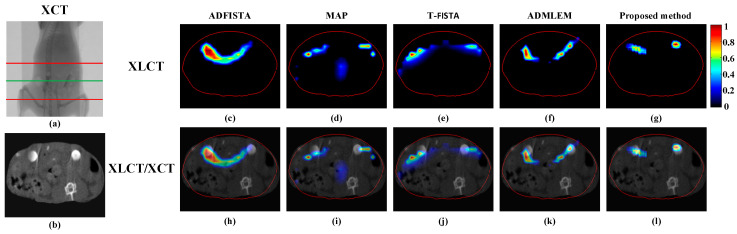
Tomographic images reconstructed based on different algorithms for mice experiments. (**a**) The CT projection image, the region between the two red lines was used for reconstruction; (**b**) the reconstructed CT slice at the height of the green line of (**a**); (**c**–**g**) reconstructions obtained using ADFISTA, MAP, T-FISTA, ADMLEM, and the proposed DeepCB-XLCT network, respectively; (**h**–**l**) the corresponding fusion results of XLCT and CT.

**Table 1 bioengineering-11-00874-t001:** *CNR* and *DICE* metrics for the reconstruction results of two targets with varying EEDs obtained from numerical simulations.

Methods	2 mm		1.5 mm		1.0 mm	
	*CNR*	*DICE*	*CNR*	*DICE*	*CNR*	*DICE*
ADFISTA	2.47	0.44	2.58	0.41	2.66	0.42
MAP	2.83	0.38	3.01	0.37	3.16	0.38
T-FISTA	1.06	0.26	2.39	0.47	2.31	0.45
ADMLEM	2.19	0.44	2.39	0.46	2.56	0.48
Proposed method	4.32	0.89	4.69	0.97	4.49	0.96

**Table 2 bioengineering-11-00874-t002:** *CNR* and *DICE* metrics for the reconstruction results of two targets with varying EEDs under different noise levels from numerical simulations.

Noise Level	2.0 mm		1.5 mm		1.0 mm	
	*CNR*	*DICE*	*CNR*	*DICE*	*CNR*	*DICE*
30 dB	4.32	0.89	4.69	0.97	4.49	0.96
25 dB	4.08	0.81	4.49	0.95	3.94	0.38
20 dB	3.60	0.78	4.04	0.90	3.76	0.45
15 dB	2.93	0.69	3.39	0.80	3.24	0.81

**Table 3 bioengineering-11-00874-t003:** *CNR* and *DICE* metrics for the reconstruction results of two targets with varying EEDs in the ablation experiment.

Methods	2.0 mm		1.5 mm		1.0 mm	
	*CNR*	*DICE*	*CNR*	*DICE*	*CNR*	*DICE*
Without skip connection and ROILoss	1.99	0.47	2.19	0.58	2.20	0.57
Without skip connection	4.05	0.86	4.47	0.96	4.19	0.94
Without ROILoss	3.31	0.85	3.69	0.90	3.55	0.91
Proposed method	4.32	0.89	4.69	0.97	4.49	0.96

**Table 4 bioengineering-11-00874-t004:** *CNR* and *DICE* metrics for the reconstruction results of two targets with varying EEDs in phantom experiments.

Methods	2.3 mm		1.7 mm		1.0 mm	
	* CNR *	* DICE *	* CNR *	* DICE *	* CNR *	* DICE *
ADFISTA	3.94	0.56	4.03	0.48	4.76	0.41
MAP	3.62	0.49	4.88	0.56	4.36	0.45
T-FISTA	3.41	0.57	7.64	0.73	3.86	0.53
ADMLEM	7.64	0.75	7.92	0.78	4.86	0.58
Proposed method	7.57	0.56	11.09	0.79	12.06	0.71

**Table 5 bioengineering-11-00874-t005:** *CNR* and *DICE* metrics for the reconstruction results of two targets in mouse experiments.

	ADFISTA	MAP	T-FISTA	ADMLEM	Proposed Method
*CNR*	3.15	1.61	2.55	3.63	4.33
*DICE*	0.32	0.16	0.24	0.35	0.60

**Table 6 bioengineering-11-00874-t006:** Reconstruction time with different methods in mice experiments.

Methods	ADFISTA	MAP	T-FISTA	ADMLEM	Proposed Method
Reconstruction time/s	14.9	48.1	48.6	15.8	1

## Data Availability

Data underlying the results presented in this paper are not publicly available at this time but may be obtained from the authors upon reasonable request.

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
