# Peer review of "Dual and Multi-Target Cone-Beam X-ray Luminescence Computed Tomography Based on the DeepCB-XLCT Network"

_bioengineering, 2024, doi:10.3390/bioengineering11090874_

Round 1
Reviewer 1 Report
Comments and Suggestions for Authors
Liu et al presented a deep-learning based tomographic reconstruction workflow for a prototype of a combined X-ray computed tomography / X-ray luminescence system (XLCT). Unlike conventional reconstruction methods based on iterative optimization of a physics model-based cost function with regularization terms, the authors proposed a 3D Encoder-Decoder network comprising several layers of spatial convolution and deconvolution, directly mapping the distribution of fluorescent sources with superficially detected fluorescence light. Both simulation and real experiments have been performed, showing promising details as compared with previous models. I found that the article fits well in the journal scope, and presents useful and original results.
I detected several minor issues in the paper, as listed below, but besides this I wonder if there was any evaluation on the radiation burden in mice due to the X-ray fluorescence acquisition. As far as I understand, CT and XLCT are done in two consecutive scans, even though some of the conventional X-ray CT projections can be used for fluorescence imaging as well. I recomment the authors to justify this choice regarding the experimental setup.
To summarize, these are my recommended amendments:
1) Experimental setup and utilization of CT data on XLCT experiments: what is the additional time-current product (mAs) required for your experimental XLCT measurement, as compared to the structural CT scan? Can fluorescence images just be acquired along with normal X-ray projections avoiding unjustified additional radiation burden to the mouse? This is a relevant topic for further utilization of this novel imaging modality on real research scenarios.
2) Please add some more details on the time employed to train your model with your current hardware configuration, and let other researcher understand if it's feasible to do the same even without those 2 fancy NVIDIA A100 on their own labs. What about reconstruction times with your new method and the established ones?
3) (from here on is just a list of typos and minor corrections) Page 3, line 97, "emitted light". Please change to "emitted light from point r".
4) Page 3, eq 5 and line 119: y and W. These are vector, please ensure uniformity of notation and use bold character. This shjould be verified throughout the entire manuscript.
5) Page 3, eq 6: the error term should be ||Wx-y||, the b vector is not defined here.
6) Page 3, line 126: "L2 and L1 regularization, respectively". Did you mean L1 and L2, respectively?
7) Page 3, line 137: "... proposed in this work are shown..." -> change to -> "... is shown ..."
8) Page 3, line 140: "y and x". Same as point 3 above (these are vectors, use bold characters).
9) Page 4, line 160, caption of Figure 1: "unsampling" -> "upsampling"
10) Page 4, line 168. Please add some reference to articles using FEM methods with detailed explanation for this precise step.
11) Page 6, line 132. Learning rate: please use scientific notation (2x10^-5 or 2E-5)
12) Page 8, lines 296-297. X-ray CT experiment. Please confirm that these projections are additional to any other projection on the same angle acquired for XLCT purposes (refer to point 1 above).
13) Page 8, line 298. "Anatomical structure" -> better saying "to reconstruct the conventional CT image" (of the tubes for validation). Similarly, at page 13, line 428, I would avoid "anatomical" in this context, rather speaking about "localization and structure of the tubes".
Comments on the Quality of English LanguageI just warned the authors about the use of the word "anatomical" in the context of phantom experiments. But grammar and style was good almost everywhere.
Author Response
Thank you for your comments and suggestions. In the revised version, we have tried our best to address all your concerns and hope the responses and revision would be satisfactory.
Please see the attachment for the point-to-point responses to your recommended amendments.

Reviewer 2 Report
Comments and Suggestions for Authors
1. I would also include the following keywords in the list: “X-ray radiation”, “X-ray luminescence (fluorescence)”, “contrast (contrast sensitivity)”, “spatial resolution”.
2. Throughout the text, a space must be placed before the opening square bracket “[”.
3. Check formula (6), perhaps the first term needs a subscript b?
4. Line 110, there must be a reference to formula (3).
5. Line 122, perhaps a reference to equation (3).
6. Line 127. The variable N must be written in italics.
7. In all figures, the dimension must be separated from the value by a space, for example, “1 mm”.
8. Explanations for formula (12), vectors must be typed in the same font as in formula (12).
9. References must be formatted uniformly. References are formatted in MDPI style. A link to https://doi.org/... makes it easier for the curious reader to access the cited article.
Wang, S.; Xia, X.; Ye, L.; Yang, B. Automatic Detection and Classification of Steel Surface Defect Using Deep Convolutional Neural Networks. Metals 2021, 11, 388. https://doi.org/10.3390/met11030388
Please be careful when correcting the text of the manuscript, perhaps I have missed something else.

Author Response

(The authors gave the same response as above.)
